# Mutations in the efflux pump regulator MexZ shift tissue colonization by *Pseudomonas aeruginosa* to a state of antibiotic tolerance

Pablo Laborda [1,2] ✉, Signe Lolle[1], Sara Hernando-Amado [3], Manuel Alcalde-Rico [3,4,5], Kasper Aanæs[6], José Luis Martínez [3], Søren Molin[2] & Helle Krogh Johansen [1,7] ✉

Mutations in *mexZ*, encoding a negative regulator of the expression of the *mexXY* efflux pump genes, are frequently acquired by *Pseudomonas aeruginosa* at early stages of lung infection. Although traditionally related to resistance to the first-line drug tobramycin, *mexZ* mutations are associated with low-level aminoglycoside resistance when determined in the laboratory, suggesting that their selection during infection may not be necessarily, or only, related to tobramycin therapy. Here, we show that *mexZ*-mutated bacteria tend to accumulate inside the epithelial barrier of a human airway infection model, thus colonising the epithelium while being protected against diverse antibiotics. This phenotype is mediated by overexpression of *lecA*, a quorum sensing-controlled gene, encoding a lectin involved in *P. aeruginosa* tissue invasiveness. We find that *lecA* overexpression is caused by a disrupted equilibrium between the overproduced MexXY and another efflux pump, MexAB, which extrudes quorum sensing signals. Our results indicate that *mexZ* mutations affect the expression of quorum sensing-regulated pathways, thus promoting tissue invasiveness and protecting bacteria from the action of antibiotics within patients, something unnoticeable using standard laboratory tests.

The increase of infections caused by antibiotic-resistant bacteria with limited therapeutic options is recognized as one of the major health challenges we are currently facing[1–3]. Indeed, almost 5 million deaths have been estimated to be associated with antimicrobial resistance (AMR) worldwide only in 2019[2]. Among antibiotic-resistant bacteria, the World Health Organisation has highlighted *Pseudomonas aeruginosa* as a microorganism with high clinical relevance and high-risk associated AMR[4,5]. It is considered one of the most difficult bacteria to eradicate due to its low susceptibility to a wide range of antibiotics and its overwhelming capacity to acquire AMR, mainly by mutations,

during extended periods of chronic infections[6–10]. As a human pathogen, *P. aeruginosa* stands out for its capacity to produce nosocomial infections and to establish chronic lung infections. These can persist for decades in Cystic Fibrosis (CF) patients, often being responsible for severe morbidity and mortality for these patients[11,12].

Among the elements causing the concerning AMR phenotype of *P. aeruginosa*, multidrug efflux pumps stand out. They are protein complexes located in the bacterial membrane that extrude different types of antibiotics outside the cell. Through this detoxification activity, efflux pumps have a role in intrinsic AMR. Besides, they can

[1]Department of Clinical Microbiology 9301, Rigshospitalet, Copenhagen, Denmark. [2]The Novo Nordisk Foundation Center for Biosustainability, Technical University of Denmark, Kgs. Lyngby, Denmark. [3]Centro Nacional de Biotecnología, CSIC, Madrid, Spain. [4]Instituto de Biomedicina de Sevilla, Hospital Universitario Virgen Macarena, CSIC, Universidad de Sevilla, Sevilla, Spain. [5]Centro de Investigación Biomédica en Red de Enfermedades Infecciosas, Instituto de Salud Carlos III, Madrid, Spain. [6]Department of Otorhinolaryngology, Head and Neck Surgery & Audiology, Rigshospitalet, Copenhagen, Denmark. [7]Department of Clinical Medicine, Faculty of Health and Medical Sciences, University of Copenhagen, Copenhagen, Denmark. ✉e-mail: palama@biosustain.dtu.dk; hkj@biosustain.dtu.dk

also contribute to acquired resistance when mutants overproducing these elements are selected[13–15]. In addition, efflux pumps are able to extrude various types of compounds, having a relevant role in several aspects of bacterial physiology and homeostasis. Some of these extruded compounds are Quorum Sensing (QS) signalling molecules or their precursors[16], which regulate the production of virulence components[17]. Therefore, efflux pumps have also a potential role in bacterial virulence regulation by modifying the intracellular concentration of QS signals. Particularly, the efflux pumps MexAB-OprM, MexCD-OprJ and MexEF-OprN export QS-related molecules and several types of antimicrobials. Hence, *P. aeruginosa* resistant mutants overproducing these systems are on one hand resistant to different antibiotics and on the other hand display a modified virulence potential due to changes in motility or secretion of toxins and tissue-damaging proteases[18–21].

Concerning its relevance in clinical settings, MexXY-OprM is one of the most remarkable *P. aeruginosa* efflux pumps. Its most prevalent substrates are aminoglycoside antibiotics such as tobramycin[22], one of the first-line drugs used to treat *P. aeruginosa* infections[23,24]. Indeed, mutations in the gene encoding the negative regulator of the expression of *mexXY*, *mexZ*, leading to *mexXY* overexpression, have been reported as being among the most frequently acquired mutations during chronic infections of *P. aeruginosa* within human airways[25–27]. Although traditionally related to tobramycin resistance[25,28], mutations in *mexZ* actually contribute just to low levels (non-clinically relevant in most cases) of AMR in in vitro standard antibiotic susceptibility testing[29]. Moreover, adaptive laboratory evolution experiments in the presence of tobramycin, including different growing conditions, rarely produce the selection of mutations in *mexZ*[30–32]. These results suggest that the driving force of the accumulation of *mexZ* mutations during *P. aeruginosa* infections is not (or not only) related to AMR.

Therefore, here we investigated the colonization strategy of a *mexZ P. aeruginosa* mutant strain in a human airway epithelium infection model. We observed that mutations in *mexZ* introduce diverse changes in the infection process in comparison with the wild-type ancestor, which may explain the overwhelmingly high frequency of these mutants in clinics. Our results point to a potential mechanism for antibiotic treatment failure rooted in modified host-microbe interaction dynamics. In addition, we highlight that, in order to improve the management of bacterial infections, we need to fully understand the extent to which antibiotic eradication is conditioned by the bacterial interaction with the relevant host tissue.

## Results

### The colonization process of a *mexZ* mutant in a human airway infection model

In order to search for additional phenotypic consequences of *mexZ* mutations during *P. aeruginosa* infection beyond causing low levels of AMR, we constructed a loss-of-function mutant in *mexZ* (*mexZ*\*; see Supplementary Table 1) in a laboratory PAO1 wild-type strain (see Methods). In this study, we address the following question: if loss-of-function *mexZ* mutations increase resistance to aminoglycosides by only a factor of two or less, why are these mutations among the most frequently observed in clinical isolates of *P. aeruginosa*? To answer this question we decided to investigate isogenic pairs of strains concerning the roles of specific mutations on complex microbial behaviour. In this way, the complicating influence of multiple other mutations present in the genomes of clinical strains was avoided. Mutations in the *mexZ* gene found in *P. aeruginosa* clinical strains are diverse, but a common trend is their frequent involvement in the C-terminal region of the protein, which is needed for its dimerization[29]. We therefore decided to construct an isogenic mutant of PAO1 by introducing a C307T SNP in the gene *mexZ*, causing an early STOP codon in the position 103 and disrupting the C-terminal domain of MexZ. Importantly, this specific SNP produces the overexpression of *mexXY* (Supplementary Dataset 1).

Next, we studied the early stages of bacterial infection of the *mexZ*\* mutant and of its parental strain, in a newly established in vitro infection model of the human airway epithelium, resembling the conditions encountered by *P. aeruginosa* when establishing infections in the lungs. This Air Liquid Interface (ALI) model comprises a columnar cell epithelium formed by all the characteristic differentiated cell types normally associated with in vivo airway epithelium (i.e., ciliated, basal, club and secretory cells) with normal functionality[33].

Since mutations in *mexZ* are frequently found in clinical CF isolates, we first analysed the infection of the *mexZ*\* mutant in an ALI infection system derived from primary cells obtained from a person with CF. After co-culturing the bacteria and host cells for 14 h at 37 °C, we performed confocal microscopy imaging of infected epithelia, and we observed a clear difference between the wild-type PAO1 and the *mexZ*\* mutant. The *mexZ*\* bacteria were abundantly attached to and frequently accumulated inside the epithelium (Fig. 1a). In contrast, the wild-type strain formed colonies in the apical side of the epithelium causing its disruption from that compartment (Fig. 1a).

The infection of this mutant was also studied in an ALI epithelia model generated using by the BCi-NS1.1 cell line. Importantly, when performing confocal microscopy imaging of the infected epithelium, we observed a similar tendency in this alternative bacterial behavior by the *mexZ*\* mutant compared to the wild-type PAO1 previously seen in the CF ALI cultures (Fig. 1b). Working with patient-derived primary cells involves recurrent sampling, and thus inconvenience for the patients. Therefore, based on the initial infection data with the *mexZ*\* mutant showing the same bacterial behavior in ALI models derived form primary cells and BCi-NS1.1 cells, we chose to utilize this cell line to deepen into this phenotype in further experiments.

Using the BCi-NS1.1 ALI infection systems, we determined the numbers of colony forming units (CFUs) of bacteria located in each compartment of the system at the end of the infection (i.e., non-attached bacteria from the basal and apical compartments, and bacteria firmly attached to the human cells; Fig. 1c) showing that the CFUs of *mexZ*\* attached to the human cell layer were significantly higher than those from the PAO1 infections (Fig. 1d). Lactate dehydrogenase (LDH) release was measured in order to determine the damage produced by the bacterial infections on the human cells. The disruption of the tight junctions of the epithelial barrier was determined by monitoring the transepithelial electrical resistance (TEER). It was observed that, despite of the increased bacterial biomass in contact with the human cells of the *mexZ*\* variant, there was no significant increase in damage compared to the wild-type strain (Supplementary Fig. 1a). Moreover, *mexZ*\* bacteria caused no significant further disruption of the tight junctions of the epithelium (Supplementary Fig. 1b). The levels of IL-8, as an indicator of potential immune recruitment of the infected epithelium, were determined by ELISA and no significant differences between *mexZ*\* and PAO1 infections were detected (Supplementary Fig. 1c).

### The overproduction of the lectin LecA is responsible for the *mexZ*\* differential colonization strategy

In a further attempt to understand the molecular mechanism underlying the altered *mexZ*\* infection behaviour, we performed a transcriptomic analysis of this mutant and focused on those expression changes potentially responsible for the phenotype. Besides the overexpression of the genes encoding MexXY efflux pump, we also observed an increased expression, up to 4-fold, of the genes encoding the PQS synthesis pathway enzymes, i.e., *pqsABCDE* and *phnAB* (Supplementary Dataset 1). This pathway is key in *P. aeruginosa* QS signalling, and it is involved in the regulation of different virulence factor encoding genes. *lecA*, encoding the lectin LecA, is a PQS-regulated gene[34,35], and, in our analysis, was observed overexpressed 2.43-fold in *mexZ*\* bacteria, likely because of the overexpressed PQS synthesis

a

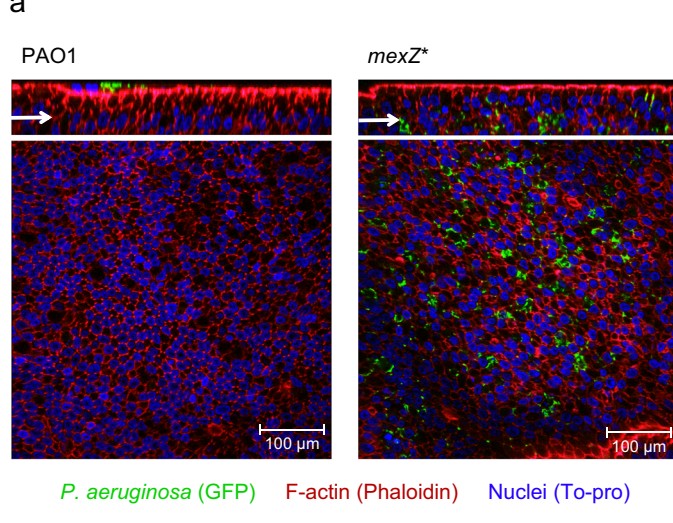

PAO1 *mexZ**

*P. aeruginosa* (GFP) F-actin (Phaloidin) Nuclei (To-pro)

b

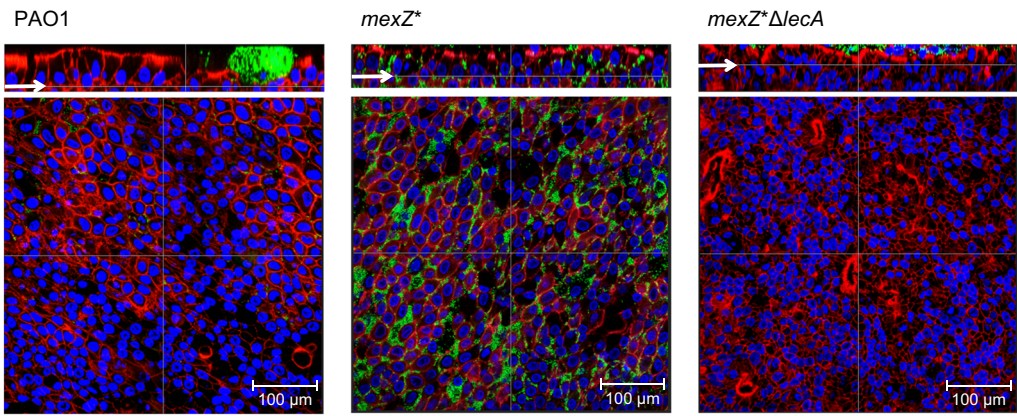

PAO1 *mexZ** *mexZ*ΔlecA*

*P. aeruginosa* (GFP) F-actin (Phaloidin) Nuclei (To-pro)

c d

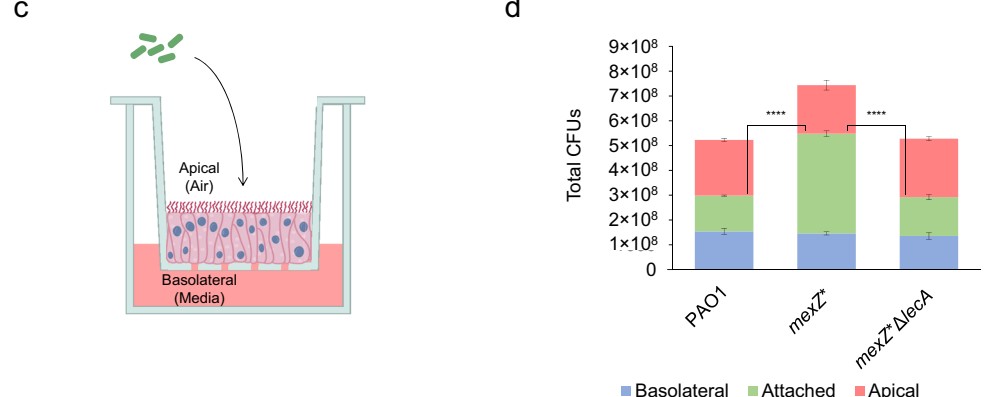

pathway encoding genes (Supplementary Dataset 1). To further confirm the transcriptomic data, *pqsD* (as representative of the PQS operon) and *lecA* expression was measured by RT-qPCR (Fig. 2). As shown, the level of *pqsD* and *lecA* expression was significantly higher in the *mexZ** mutant than in the wild-type strain. Noteworthy, LecA has been described to be involved in the invasiveness of *P. aeruginosa* towards the human epithelium and adhesion to human cells[36,37]. We

then hypothesized that its overproduction in the *mexZ** mutant might be involved in the altered infection behaviour of this mutant.

In order to address this possibility, we deleted the *lecA* gene in the *mexZ** mutant and characterised its infection behaviour. We observed that the *mexZ*ΔlecA* mutant reverted to the wild-type phenotype, both when determining the numbers of attached bacteria, and when performing confocal microscopy of infected ALI cultures (Figs. 1b, d).

**Fig. 1 | Characterization of the effects of a mutation in *mexZ* for bacterial localization during colonization of Air Liquid Interface (ALI) airway infection models. a** Confocal images of the internal part of ALI cultures derived from primary cells of a person with Cystic Fibrosis after 14 h of infection with PAO1 or *mexZ*\* *P. aeruginosa* in green (GFP), epithelial structure visualized by F-actin staining in red (Phalloidin) and nuclei in blue (To-pro), and their corresponding cross sections. Arrows in the cross section highlight the layer shown in the internal part image. Scale bar = 100 μm. The results shown were consistently obtained in three independent biological replicates of the experiment. **b** Confocal images of the internal part of fully differentiated BCi-NS1.1 cells ALI cultures after 14 h of infection with PAO1, *mexZ*\* or *mexZ*\*Δ*lecA P. aeruginosa* in green (GFP), epithelial structure visualized by F-actin staining in red (Phalloidin) and nuclei in blue (To-pro), and their corresponding cross sections. Arrows in the cross section highlight the layer shown in the internal part image. Scale bar = 100 μm. The results shown were consistently obtained in three independent biological replicates of the experiment.

**c** Schematic representation of the ALI airway infection model set up. A pseudostratified layer of differentiated epithelial cells in contact with the air from its apical side and with medium from its basolateral side through the pore membrane of the transwell in which the human cells are located, is represented. When starting an infection to be monitored, *P. aeruginosa* cells (represented by green rods) are added to the apical side of the human cell culture. **d** Colony Forming Units (CFUs) of PAO1, *mexZ*\* and *mexZ*\*Δ*lecA* in the apical (red) and basolateral (blue) ALI compartments, and attached to the cell layer (green), after 14 h of infection in fully differentiated BCi-NS1.1 cells. Data are presented as mean values ± SEM of the results from three biological replicates with three technical replicates each. Statistical significance was determined by two-sided *t*-test assuming equal variances for CFU measurements and indicated as **** (*p* ≤ 0.00005): attached CFUs PAO1 vs *mexZ*\* *p* = 0.00002; Attached CFUs *mexZ*\* Δ*lecA* vs *mexZ*\* *p* = 0.00005. Source data are provided as a Source Data file.

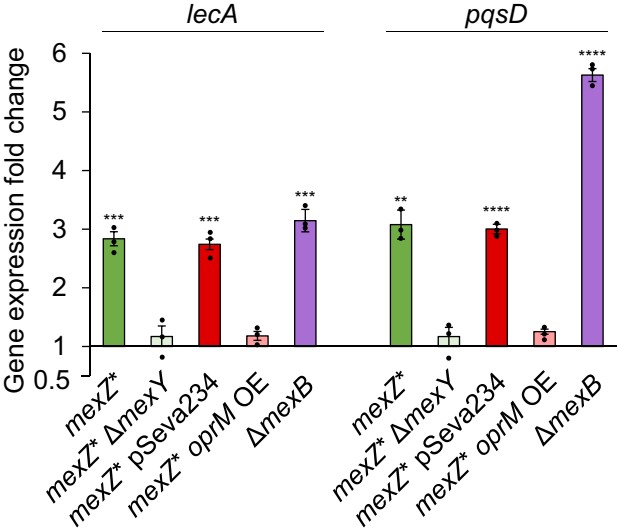

**Fig. 2 | Influence of *mexY* and *oprM* in the expression levels of *lecA* and *pqs* operon of *mexZ*\* mutant.** Fold changes of *mexZ*\* (green), *mexZ*\*Δ*mexY* (light green), *mexZ*\* overexpressing (OE) *oprM* and its control with empty pSeva234 plasmid (light red and dark red, respectively) and Δ*mexB* (purple) were estimated regarding the expression of the PAO1 wild-type strain by RT-qPCR. Data are presented as mean values ± SEM of the results from three biological replicates with three technical replicates each. Statistically significant differences regarding PAO1 were calculated with two-sided *t*-test for paired samples assuming equal variances and indicated as ** (*p* < 0.005), *** (*p* < 0.0005) and **** (*p* < 0.00005): *lecA* expression in *mexZ*\* (*p* = 0.00012), *lecA* expression in *mexZ*\*Δ*mexY* (*p* = 0.4), *lecA* expression in *mexZ*\*pSEVA234 (*p* = 0.000056), *lecA* expression in *mexZ*\**oprM* OE (*p* = 0.098), *lecA* expression in Δ*mexB* (*p* = 0.00038); *pqsD* expression in *mexZ*\* (*p* = 0.0011), *pqsD* expression in *mexZ*\*Δ*mexY* (*p* = 0.34), *pqsD* expression in *mexZ*\*pSEVA234 (*p* = 0.000017), *pqsD* expression in *mexZ*\**oprM* OE (*p* = 0.054), *pqsD* expression in Δ*mexB* (*p* = 0.000002). Source data are provided as a Source Data file.

This supports that the overexpression of *lecA* in *mexZ*\* indeed is responsible for its accumulation inside the epithelial barrier.

**The competition between MexXY-OprM and MexAB-OprM plays a central role in *lecA* overexpression in *mexZ*\***

The following steps were directed towards a deeper understanding of the mechanism by which the *mexZ*\* mutant exhibited an overexpression of *lecA*. First, we tested whether these expression changes were due to the overproduction of the MexXY-OprM efflux pump, or to a previously undescribed regulatory capacity of MexZ towards the expression of some virulence genes. To do so, we deleted the *mexY* gene in the *mexZ*\* mutant and observed that the loss of the MexXY efflux pump recovered *pqsD* and *lecA* expression levels to those of the

wild-type strain (Fig. 2). This indicates that these expression changes were directly dependent on the overproduction of the MexXY-OprM efflux pump.

Although some efflux pumps of *P. aeruginosa* have been described to have a role in the extrusion of QS signals and their precursors, this has never been observed for MexXY-OprM efflux pump. However, the role of MexAB-OprM in this aspect of bacterial physiology has been proven and studied in detail[19,21]. Indeed, we observed many changes in the expression of genes related to QS in a transcriptome of a MexAB-OprM defective mutant, Δ*mexB* (Supplementary Table 1 and Supplementary Dataset 1). Interestingly, many of these changes were coincident with those observed in the *mexZ*\* mutant: an overexpression of the PQS signalling pathway encoding genes and of *lecA* (Supplementary Dataset 1 and Supplementary Dataset 2). To further confirm these results, *pqsD* and *lecA* expression levels were determined in the Δ*mexB* mutant by RT-qPCR, which were observed significantly higher than in the wild-type strain (Fig. 2).

We then addressed how the overproduction of MexXY-OprM efflux pump and the loss-of-function of the MexAB-OprM efflux pump could be causing these parallel expression changes. Both efflux pumps share the OprM porin, for which the encoding gene is located in the *mexAB* operon[22]. Hence, the competition between MexXY and MexAB for OprM in the *mexZ*\* mutant could be in the basis for the similar expression changes observed in the *mexB* defective mutant. The non-physiological increased amount of MexXY, which requires the OprM porin to be active, could be "sequestering" this porin which, on the other hand, would stop constituting part of the MexAB-OprM efflux pump in the *mexZ*\* mutant, thus altering the extrusion of QS signals by MexAB-OprM. The accumulation of QS signals due to a reduced amount of the complete structure of MexAB-OprM, caused by the overproduction of MexXY-OprM (*mexZ*\* mutant) or by the absence of the genes encoding MexAB-OprM (Δ*mexB* mutant), would lead to the overexpression of *lecA*.

To further test if the competition between MexXY and MexAB for OprM could be in the basis of the differential tissue invasiveness observed in the *mexZ*\* mutant, we overexpressed *oprM* in the *mexZ*\* mutant (see Methods), generating a sufficient amount of OprM to cover the requirements of the overproduced MexXY efflux pump and of the basal amount of MexAB. First, we measured the expression levels of *pqsD* and *lecA* in the *mexZ*\* mutant overexpressing *oprM*, finding that they reverted back to those of the wild-type (Fig. 2). Moreover, we inspected the colonization behaviour of the Δ*mexB* mutant in the ALI infection system, finding that it displayed a phenotype similar to that observed in *mexZ*\*, accumulating inside the epithelial barrier (Supplementary Fig. 2).

All these results support our hypothesis and suggest that the interaction between MexXY and MexAB, by the competition towards its shared porin, is in fact the basis of the *mexZ*\* mutant behaviour (Supplementary Fig. 3).

### The *mexZ** differential colonization strategy is associated with an increased protection from diverse antibiotics

After observing the altered infection behaviour of the *mexZ** mutant, we wondered if it could constitute an advantage for the bacteria when colonizing the lung. In fact, many antibiotics do not easily penetrate the epithelial barrier, so it could constitute a protective environment for the bacteria to accumulate in that compartment. To address this possibility, we simulated antibiotic treatments of epithelia infected with PAO1, *mexZ** and *mexZ*ΔlecA*. The antibiotics chosen for this set up were tobramycin, ceftazidime and ciprofloxacin, which are among the most frequently used antibiotics against *P. aeruginosa* infections[23,38]. Tobramycin and ciprofloxacin are extruded by the MexXY-OprM efflux pump, although this overproduction does not lead to high in vitro levels of resistance to these two drugs[29]. In contrast, ceftazidime is not extruded by this efflux pump[22,39]. Importantly, ciprofloxacin has been described to be able to very efficiently cross the epithelial barrier, unlike tobramycin and ceftazidime[40].

First, we co-cultured bacteria of the respective strains with ALI cultures for 7 h. This leads bacteria to colonise the epithelium and to accumulate in the respective different compartments (Fig. 3a), without disrupting the epithelium (Fig. 3b). At this point, we added the respective antibiotics at the appropriate concentrations (see Methods) and measured bacterial CFUs and epithelial TEER after 12 h. The results show that tobramycin and ceftazidime were not able to effectively stop *mexZ** infection, whereas they efficiently prevented continuation of PAO1 and *mexZ*ΔlecA* infections (Figs. 3b, c). This suggests that the *mexZ** mutant colonizing the human epithelium is protected from the action of the antibiotics tobramycin and ceftazidime, which do not easily cross the human epithelium, and that this protection is due to its altered localization caused by *lecA* overexpression. Interestingly, despite the efficient penetration of ciprofloxacin into the epithelium, the *mexZ** bacteria were more protected than PAO1 from this antibiotic as was also the *mexZ*ΔlecA* strain (Figs. 3b, c). This latter finding suggests that the greater capacity to continue an infection by *mexZ** in presence of ciprofloxacin, an antibiotic efficiently reaching the internal part of the epithelium, is most likely due to the capacity of MexXY-OprM to extrude this antibiotic, and not because of its altered infection behaviour.

Finally, we determined the Minimal Inhibitory Concentrations (MIC) to these antibiotics for all the tested strains (Supplementary Table 2) by the broth dilution method in Müller Hinton (MH), the technique commonly used to determine AMR of clinical strains[41]. We measured the CFUs of each strain present in the concentration corresponding to half of the MIC of PAO1, so we had the same measured parameter as in the infection set up to compare with. We found that *mexZ** is more resistant than PAO1 to ciprofloxacin and tobramycin in laboratory media (Fig. 3d), due to their extrusion by MexXY-OprM, as previously described[29]. However, no difference in resistance to ceftazidime, which is not extruded by this efflux pump, was observed (Fig. 3d). As expected, the deletion of *lecA* did not have any effect on the AMR phenotype of *mexZ** (Fig. 3d), since the lectin LecA is not involved in AMR in in vitro MIC tests.

We also measured the MICs in Pneumacult-ALI maintenance medium and Synthetic Cystic Fibrosis sputum Medium (SCFM). The results showed that the mutation in *mexZ* slightly increases resistance to tobramycin and ciprofloxacin in all media, and that deleting *lecA* does not change the level of increased antibiotic resistance (Supplementary Table 2). Although the MIC measured is dependent on the nutrient availability and media composition, as previously described[32], the differences between mutant and wild-type strains remain consistent.

Overall, these results support that *mexZ** is efficiently protected towards diverse antibiotics when colonizing the human epithelium due to its altered tissue colonization behaviour, something not observable in common in vitro AMR determination assays. In other words, this mutant may be "hiding" from antibiotics during the colonization of the human lung epithelium, acquiring a higher protection against these drugs than expected based on MICs.

## Discussion

Antibiotic treatment of bacterial infections is increasingly complicated by resistance development, and surveillance of AMR is therefore an important diagnostic factor in the clinic. In particular, treatment of long-term infected patients requires parallel monitoring of resistance development, which is routinely performed as standardized antibiotic susceptibility assays in the clinical microbiology laboratory. The underlying assumption is that antibiotic susceptibility is an absolute trait, which can reliably be determined in laboratory settings using simple plate or tube tests.

The observations presented in this work clearly document that antibiotic susceptibility is highly conditional, and that a high proportion of bacterial variants in infected patients may in fact escape antibiotic eradication by mechanisms, which are not addressed in the clinic. The consequence may frequently be (1) that the continued treatment has limited effect, and (2) that true resistance develops over time.

We previously showed that among close to 500 *P. aeruginosa* isolates obtained from young, infected CF patients, about 40% carried mutations in the MexXY efflux pump regulator, MexZ[25]. A further analysis of these isolates, showed that only few of them were clinically relevant resistant to tobramycin[29], regularly used for treating *P. aeruginosa* infections. These observations triggered this investigation of a possible avoidance of antibiotic treatments by *mexZ* mutants, something that could explain their frequent selection in the infected patient. First, we present evidence supporting that bacterial tissue invasion is induced by the inactivation of MexZ, and next we resolve the underlying molecular mechanism behind this behaviour. Finally, we show that tissue invasion may provide protection of the bacteria from several antibiotics.

During chronic lung infections, *P. aeruginosa* acquires AMR mainly by mutations leading to a reduced antibiotic activity. Among the most frequently acquired mutations, we find those located in genes encoding the target of antibiotics, antibiotic-entrance channels or regulators of the expression of active AMR determinants (i.e., multidrug efflux pumps or β-lactamases)[25,42]. However, AMR determinants can additionally be connected to regulatory networks involved in different aspects of bacterial physiology beyond AMR, including virulence-related traits[16,43]. Hence, the acquisition of mutations affecting AMR determinants might involve additional changes in the bacterial physiology that are currently underestimated and that, in fact, may be of huge relevance during the infection process, something for which this work is a proof of concept. Specifically, we addressed the case of mutants in *mexZ* for which the overproduction of the MexXY-OprM multidrug efflux pump leads to an AMR level in in vitro tests that is scarce[29]. This observation indicated that the mutations in this specific gene may result in additional phenotypes, which might be relevant in the specific conditions of the infection site, different from those typically found during standard laboratory settings.

In this work, we confirmed this initial theory by characterizing the altered infection behaviour of a *mexZ** mutant. This mutant tends to accumulate inside the epithelial barrier during early colonization, a compartment in which a more protective environment towards diverse antibiotics from different structural families was encountered (Fig. 4). The observed phenotype was caused by a PQS signalling alteration that leads to the overexpression of *lecA*, a gene encoding a lectin crucial for the attachment of *P. aeruginosa* to human tissues[36,37,44]. This differential colonization strategy, and its potential boosting of bacterial recalcitrance, may account for the not well explained overwhelming frequency of *mexZ* mutations seen in the clinic.

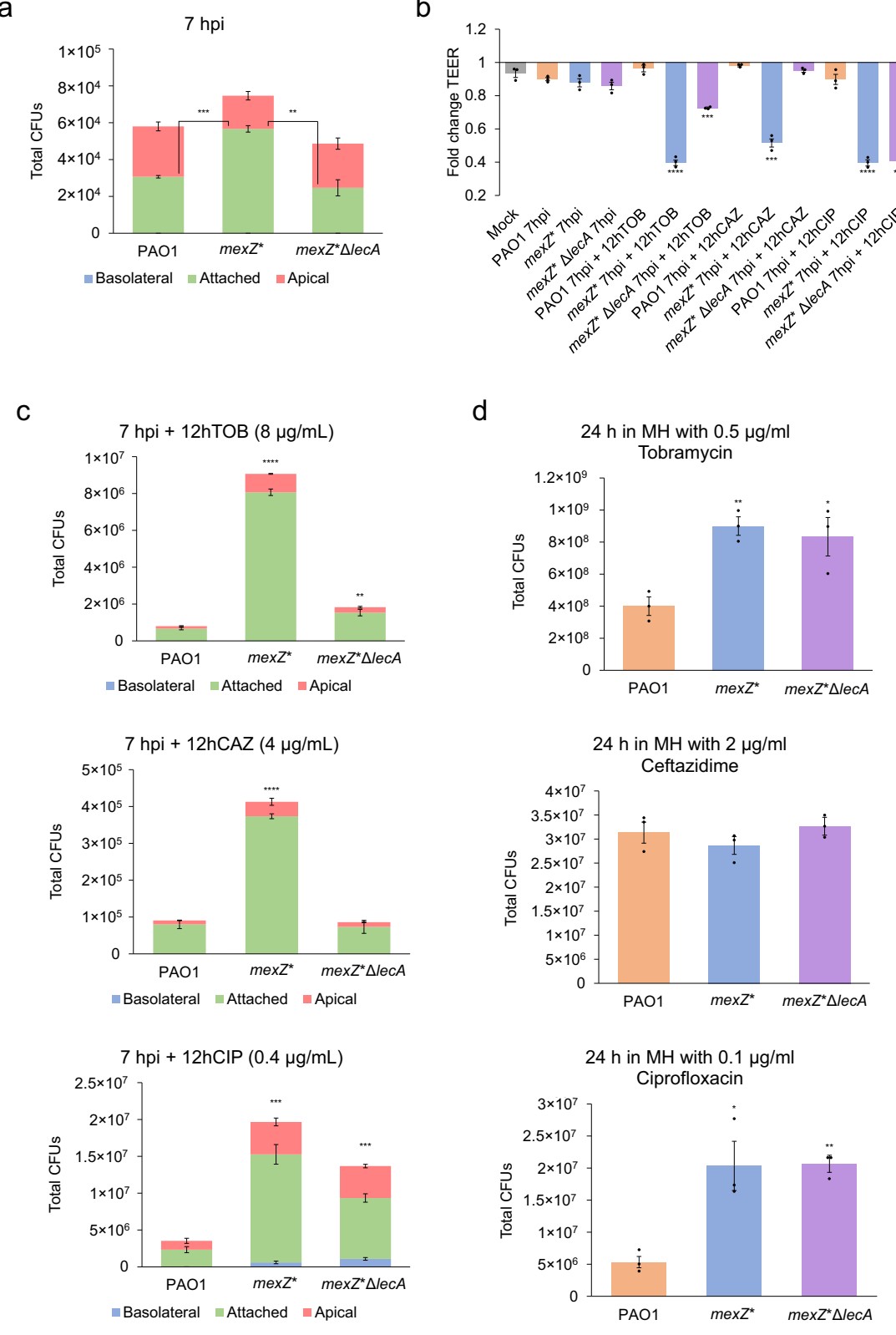

We further observed that ciprofloxacin, tobramycin and ceftazidime, the most commonly used antibiotics against *P. aeruginosa* infections[23,38], are not able to compromise a *mexZ** mutant infection as compared to a PAO1 wild-type one. Since the MexXY-OprM efflux pump is able to extrude tobramycin, its overproduction in *mexZ** and in *mexZ*ΔlecA* may contribute to tobramycin resistance, as is indicated by the fact that both mutants grow significantly more than PAO1 after

treatment with tobramycin (Fig. 3c). Nevertheless, by comparing the infection outcome after tobramycin treatment of both mutants it is observed that most of the tobramycin protection seems to come from the differential colonization strategy (associated with *lecA* over-expression). The fact that a similar protection was observed for ceftazidime (Fig. 3c), an antibiotic that is not a substrate of MexXY-OprM[22,39], reinforces the hypothesis that intercellular colonization

**Fig. 3 | Determination of the protection from antibiotic treatments due to the bacterial behaviour during infection caused by a mutation in *mexZ*. a** Colony Forming Units (CFUs) of PAO1, *mexZ**, and *mexZ**Δ*lecA* in the apical (red) and basolateral (blue) ALI compartments, and attached to the cell layer (green), after 7 h of infection in fully differentiated BCi-NS1.1 cells, a timepoint after which antibiotics were added. Data are presented as mean values ± SEM of the results from three biological replicates with three technical replicates each. Statistical significance was determined by two-sided *t*-test assuming equal variances for CFU measurements and indicated as ** ($p < 0.005$) and *** ($p < 0.0005$): attached CFUs PAO1 vs *mexZ** $p = 0.00007$; Attached CFUs *mexZ** Δ*lecA* vs *mexZ** $p = 0.0025$. **b** Fold change of Transepithelial Electrical Resistance (TEER) (Ω·cm²) of mock ALI cell layers uninfected (grey) and after 7 h of infection or 7 h of infection followed by 12 h of treatment with tobramycin (TOB), ceftazidime (CAZ) of ciprofloxacin (CIP) of PAO1 (orange), *mexZ** (blue) or *mexZ**Δ*lecA* (purple) strains respect to the TEER before starting the infection experiment. Data are presented as mean values ± SEM of the results from three biological replicates. Statistical significance with respect to the mock uninfected control cells was determined by two-sided *t*-test assuming equal variances for TEER measurements and indicated as *** ($p < 0.0005$) and **** ($p < 0.00005$): PAO1 7hpi ($p = 0.19$), *mexZ** 7 hpi ($p = 0.15$), *mexZ**Δ*lecA* 7hpi ($p = 0.06$), PAO1 7hpi + 12 h TOB ($p = 0.33$), *mexZ** 7hpi + 12 h TOB ($p = 0.000027$), *mexZ**Δ*lecA* 7hpi + 12 h TOB ($p = 0.00046$), PAO1 7hpi + 12 h CAZ ($p = 0.08$), *mexZ** 7hpi + 12 h CAZ ($p = 0.00017$), *mexZ**Δ*lecA* 7hpi + 12 h CAZ ($p = 0.51$), PAO1 7hpi + 12 h CIP ($p = 0.38$), *mexZ** 7hpi + 12 h CIP ($p = 0.000034$), *mexZ**Δ*lecA* 7hpi + 12 h

CIP ($p = 0.000051$). **c** CFUs of PAO1, *mexZ**, and *mexZ**Δ*lecA* in the apical (red) and basolateral (blue) ALI compartments, and attached to the cell layer (green), after 7 h of infection followed by 12 h of treatment with tobramycin (TOB), ceftazidime (CAZ) or ciprofloxacin (CIP) in fully differentiated BCi-NS1.1 cells. Data are presented as mean values ± SEM of the results from three biological replicates with three technical replicates each. Statistical significance for CFU measurements after antibiotic treatments in the infection system regarding PAO1 was determined by two-sided *t*-test assuming equal variances and indicated as ** ($p < 0.005$), *** ($p < 0.0005$) and **** ($p < 0.00005$): *mexZ** 7hpi + 12 h TOB ($p = 0.0000025$), *mexZ**Δ*lecA* 7hpi + 12 h TOB ($p = 0.0036$), *mexZ** 7hpi + 12 h CAZ ($p = 0.0000057$), *mexZ**Δ*lecA* 7hpi + 12 h CAZ ($p = 0.65$), *mexZ** 7hpi + 12 h CIP ($p = 0.000064$), *mexZ**Δ*lecA* 7hpi + 12 h CIP ($p = 0.00038$). **d** CFUs of PAO1 (orange), *mexZ** (blue), and *mexZ**Δ*lecA* (purple) present in the concentration corresponding to half of the PAO1 MIC of tobramycin (TOB), ceftazidime (CAZ), or ciprofloxacin (CIP) in 100 µL of Müller Hinton (MH) medium. Data are presented as mean values ± SEM of the results from three biological replicates with three technical replicates each. Statistical significance for CFU measurements in the MIC determination set up in MH regarding PAO1 was determined by two-sided *t*-test assuming equal variances and indicated as * ($p < 0.05$) and ** ($p < 0.005$): *mexZ** TOB ($p = 0.0036$), *mexZ**Δ*lecA* TOB ($p = 0.031$), *mexZ** CAZ ($p = 0.4$), *mexZ**Δ*lecA* CAZ ($p = 0.66$), *mexZ** CIP ($p = 0.019$), *mexZ**Δ*lecA* CIP ($p = 0.00066$). Source data are provided as a Source Data file.

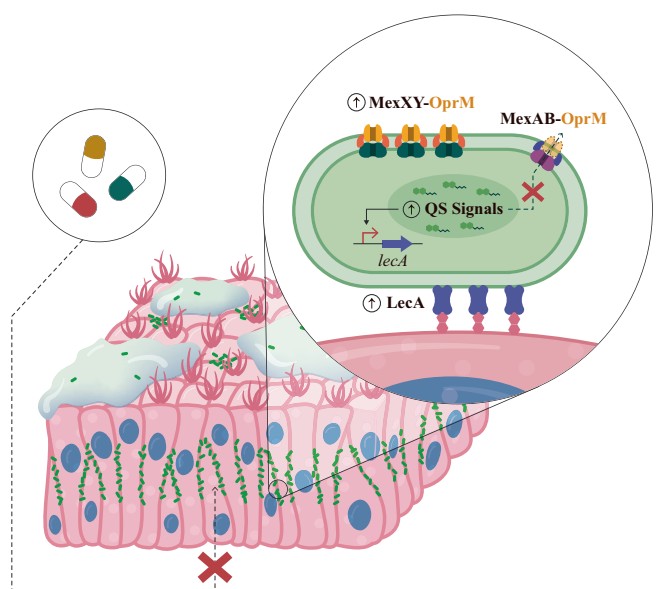

**Fig. 4 | Schematic illustration of *mexZ** colonization behaviour.** Bacteria with a loss-of-function mutation in *mexZ* are more prone to accumulate inside the epithelial barrier, a compartment in which many antibiotics (represented by pills) barely reach. The mutation in *mexZ* leads to an overproduction of MexXY-OprM efflux pump, reducing the amount of available OprM porin for the formation of MexAB-OprM efflux pump. The latter causes an alteration of the intracellular amount of Quorum Sensing (QS) signals normally extruded by MexAB-OprM. The disruption of the level of QS signals causes an overexpression of *lecA*, which encodes a lectin involved in *Pseudomonas aeruginosa* invasiveness and attachment to epithelial cells. The result is an increased protection from antibiotics during infection of lung epithelia.

could be more protective than the antibiotic efflux itself. However, the efflux by the overproduced MexXY-OprM, and not the infection behaviour, seems to be the major cause of protection of *mexZ** during infection towards ciprofloxacin (Fig. 3c, d), an antibiotic efficiently crossing the epithelial barrier. This suggests that choosing antibiotics that efficiently penetrate the epithelial barrier may be a starting point leading to an efficient therapeutic approach against this bacterial behaviour.

Our results provide a better understanding of the infection lifestyle of *P. aeruginosa*, in particular of *mexZ* mutants, which are very frequently selected for during infections of people with CF[25,45]. According to our results, this enrichment could be further boosted by the higher susceptibility of CF epithelia to invasion[46]. Our findings point to an increased recalcitrance to antibiotic treatment associated with intraepithelial invasion as a relevant driver of the selection of *mexZ* mutations during infection. Besides, whether this behaviour also provides advantages towards other challenges within the infected tissue, such as the action of immune cells or low oxygen concentrations, remains to be analyzed. Efficiently invading the intraepithelial compartment could also give access to a niche of nutrients distinct from those in the mucus layer. This is suggested by the finding that the *mexZ** mutant population was observed to be larger than the PAO1 population at the studied timepoint of infection (Fig. 1d). The impact of the recent advancements in CF patient treatment through modulator therapy[47] on the frequency of *mexZ* mutations in clinical strains of *P. aeruginosa* will be assessed in the coming years.

The results of this work also point to the competition between MexXY and MexAB, due to their shared OprM porin, as the basis of the altered QS signalling and infection behaviour of the *mexZ** mutant. The overproduced MexXY may "sequester" the OprM porin, leading to a decrease in the total amount of the active MexAB-OprM efflux pump and, consequently, in the QS signals extruded by this latter efflux pump. The QS signals and QS precursors extruded by MexAB-OprM efflux pump will thus likely accumulate inside the bacterium, leading to an altered QS signalling and an alteration of their regulated genes (Supplementary Fig. 3). Expression levels of the *mexAB-oprM* operon are similar in the *mexZ** mutant and the wild-type strain, which may account for the lack of sufficient amounts of OprM for both MexAB and the overproduced MexXY efflux pump. Moreover, when overexpressing *oprM* in *mexZ**, the expression profiles of genes related to the *mexZ** behaviour (i.e., *lecA*) reverted to wild-type levels (Fig. 2). Finally, the colonization pattern observed in a Δ*mexB* mutant was similar to that of the *mexZ** mutant (Supplementary Fig. 2), an observation that may explain the accumulation of several mutations in the MexAB encoding genes during chronic lung infections of *P. aeruginosa*, despite they result in an efflux pump deficiency[25,27,48]. Although there is a phenotypic convergence associated with *lecA* overexpression that can result from mutations in *mexZ* or *mexAB*, the first ones are more frequently found in clinical strains. The reason could be that

mutations in *mexZ*, which cause the overproduction of the MexXY-OprM efflux pump, may confer an adaptive advantage during antibiotic treatment. On the other hand, loss-of-function mutations in the genes encoding the MexAB-OprM efflux system will be counter-selected in the presence of antibiotics detoxified by this efflux pump, such as β-lactam antibiotics which are frequently used during chronic infections. Since the behaviour associated with mutations in *mexZ* or *mexAB* is caused by QS dysregulation, the co-occurrence of these mutations and mutations in QS regulators encoding genes, which is noted in some clinical strains[25], might impact this phenotype. This posibility could also present an intriguing avenue for further investigation into the described phenotype.

Overall, this work demonstrates to which extent new patterns of host-microbe interactions, due to the modification of bacterial behaviour by an altered export of signal molecules, can result in the acquisition of AMR in *P. aeruginosa*. This study can set a landmark in establishing a new scope for a deeper understanding of the threatening problem of AMR acquisition. In fact, this work provides insights into the role of AMR determinants beyond resisting the action of antimicrobials, in particular the links between AMR elements and infection processes. It sheds further light on the adaptation processes of *P. aeruginosa* in the human lung environment, focusing on the acquisition of mutations affecting AMR determinants, which has so far underestimated consequences for infections. Our results suggest that the fixation of the mutations in specific AMR determinant encoding genes within infecting bacterial populations may be driven by advantageous alterations in bacterial infection behaviour. Importantly, our findings strongly suggest that this potential collateral outcome of the acquisition of AMR should be taken into consideration for improving the diagnoses and managements of infectious diseases.

## Methods

This study was performeed following the approval of the local ethics committee of the Capital Region of Denmark (Region Hovedstaden) with registration number H-20024750. All patients signed informed consent to participate in the study and publish the associated results and relevant clinical data. Pseudoanonymisation of patient information was carried out to discontinue any connection that would allow patient identification.

### Construction of bacterial strains

All strains, plasmids and oligonucleotides used for this work appear in Supplementary Table 1.

Construction of punctual and deletion mutants of *P. aeruginosa* were performed by homologous recombination with a pEX18Ap plasmid. For the construction of the mutation in the gene *mexZ*, the corresponding pEX18Ap_*mexZ** plasmid (Supplementary Table 1) was constructed. The upstream and downstream regions of the position in which the genetic modification was intended to be introduced were amplified by PCR with primer pairs EcoRI_*mexZ**Upstream_Fw/*mexZ**Upstream_Rv and *mexZ**Downstream_Fw/BamHI_*mexZ**Downstream_Rv (Supplementary Table 1), respectively. By using the PCR products as templates for a PCR reaction with oligonucleotide pairs EcoRI_*mexZ**Upstream_Fw/BamHI_*mexZ**Downstream_Rv (Supplementary Table 1), respectively, both fragments were joined in a sequence containing the EcoRI and BamHI restriction enzyme target sequences in its ends. The resulting products were digested with EcoRI and BamHI restriction enzymes (New England Biolabs) and cloned into a HindIII-digested and dephosphorylated pEX18Ap plasmid, by using the T4 DNA ligase (Promega) following suppliers instructions[49]. For the deletion of *lecA* and *mexY* genes, the corresponding pEX18Ap_Δ*lecA* and pEX18-ap_Δ*mexY* plasmids (Supplementary Table 1) were constructed. First, the upstream and downstream regions of *lecA* or *mexY* were independently amplified using the oligonucleotide pairs HindIII_*lecA*Upstream_Fw/*lecA*Upstream_Rv and *lecA*Downstream_Fw/HindIII_*lecA*Downstream_Rv

or HindIII_*mexY*Upstream_Fw/*mexY*Upstream_Rv and *mexY*Downstream_Fw/HindIII_*mexY*Downstream_Rv (Supplementary Table 1), respectively. The PCR products were used as templates for a subsequent PCR reaction with oligonucleotide pairs HindIII_*lecA*Upstream_Fw/HindIII_*lecA*Downstream_Rv or HindIII_*mexY*Upstream_Fw/HindIII_*mexY*Downstream_Rv (Supplementary Table 1), respectively, containing the HindIII restriction enzyme target sequence and joining both fragments. The resulting products were digested, as described above, with the HindIII restriction enzyme (New England Biolabs) and cloned into the pEX18Ap plasmid as described above. After checking the sequence of the obtained plasmids by Sanger sequencing using the primers M13_Fw and M13_Rv (Supplementary Table 1), they were introduced by transformation into chemically competent *E. coli* S17-1λ pir cells[50]. These *Escherichia coli* S17-1λ pir cells were used as a donor strain towards the PAO1 wild-type *P. aeruginosa* strain, in which the intended SNP was introduced by homologous recombination. For its part, the corresponding *E. coli* S17-1λ pir containing the *lecA* or *mexY* deletion plasmid was used as a donor strain towards the *mexZ** *P. aeruginosa* mutant, in which the gene *lecA* or *mexY* was deleted by homologous recombination, respectively. The plasmids were conjugated from *E. coli* S17-1λ pir to the respective *P. aeruginosa* strain, and the *P. aeruginosa* cells containing the plasmid were selected on *Pseudomonas* Isolation Agar (PIA) plates containing the appropriate antibiotic of selection. Merodiploids were resolved by plating on PIA containing 5% of sucrose[51]. The introduction of the SNP in *mexZ* was verified by Sanger sequencing of the corresponding genetic region with the primer pair *mexZ**_comp_Fw/*mexZ**_comp_Rv (Supplementary Table 1). Deletion of *lecA* and *mexY* was verified by PCR with the primer pairs Δ*lecA*_comp_Fw/Δ*lecA*_comp_Rv and Δ*mexY*_comp_Fw/Δ*mexY*_comp_Rv (Supplementary Table 1).

### GFP-tagging of bacterial strains

The *P. aeruginosa* strains whose infection was monitored were firstly GFP-tagged to enable visualization by confocal microscopy analyses. Bacterial strains were GFP-tagged by introducing a pUC18T-mini-Tn7T-Gm-Tp-*gfp* plasmid (Supplementary Table 1) by conjugation. Conjugation was accomplished by four parental mating, by placing a drop of culture containing *E. coli* DH5α/pUC18T-mini-Tn7T-Gm-Tp-*gfp*, *E. coli* SM10(λ*pir*)/pTNS1, *E. coli* HB101/pRK2013 (Supplementary Table 1) and the respective *P. aeruginosa* strain in the same amount in a 3-mm cellulose acetate filter membrane placed on nonselective LB-agar incubated at 37 °C overnight[52]. Transformants were selected in PIA with the appropriate antibiotic of selection. Subsequently, a Flp-mediated excision of the antibiotic resistance marker was performed with a pFLP2 plasmid (Supplementary Table 1)[52]. For that, electro-competent cells of the respective *P. aeruginosa* strain resulting from conjugation were prepared by washing with sucrose 300 mM twice, the pFLP2 plasmid was introduced by electroporation and transformants were selected in PIA with the appropriate antibiotic of selection[52]. The curation of the pFLP2 plasmid was performed by plating in LB-agar plates with 5% of sucrose. The final GFP-tagged strains were verified by checking green fluorescence.

### Gene overexpression in bacterial strains

For the overexpression of *oprM*, the *oprM* gene sequence was amplified with oligonucleotides AvrII_*oprM*_Fw and HindIII_*oprM*_Rv (Supplementary Table 1) containing the restriction enzyme target sequences of AvrII and HindIII. The resulting PCR product was digested by those restriction enzymes and cloned using the T4 DNA ligase (New England Biolabs) into a pSEVA234 plasmid (Supplementary Table 1) digeted with the same enzymes[53]. Competent *mexZ** *P. aeruginosa* strain cells were prepared by washing them twice with 150 mM of $MgCl_2$ followed by a 1 h incubation in ice[54], and the pSEVA234*oprM* overexpressing plasmid (Supplementary Table 1) and pSEVA234 empty plasmid, as a control, were introduced in the bacteria by transformation.

## Air Liquid Interface infection model preparation

BCi-NS1.1 cells[55] were used for the generation of ALI infection model systems. Cells were initially expanded in culture flasks in a humidified incubator at 5% $CO_2$ and 37 °C, using Pneumacult-Ex Plus medium (STEMCELL Technologies) following manufacturer's instructions. After expansion, $10^5$ cells were seeded onto 1 μm pore polyester membrane inserts (Falcon) previously coated with type I collagen (Gibco). Once cells reached full confluency, ALI conditions were established by removing media from the apical chamber and replacing the media of the basolateral chamber with Pneumacult-ALI maintenance medium (STEMCELL Technologies) supplemented with 4 μg/mL heparin (STEMCELL Technologies), 480 ng/mL hydrocortisone, Pneumacult-ALI 10x supplement (STEMCELL Technologies) and Pneumacult-ALI maintenance supplement (STEMCELL Technologies). ALI cultures were grown at 37 °C and 5% $CO_2$ in humidified incubator for 28 days. Every 3–4 days, media was replaced. TEER was measured regularly using a chopstick electrode (STX2; World Precision Instruments) for monitoring epithelial polarization. After the initial 15 days under ALI conditions, the accumulated mucus was removed by washing apical surface with PBS every 7 days.

For establishment of CF ALI cultures, nasal mucosa was obtained from a person with CF undergoing endoscopic endonasal sinus surgery. Tissue specimens were surgically removed and immediately placed in Dulbecco´s modified essential media (DMEM) supplemented with 600 μg/mL penicillin and 1 mg/mL streptomycin until further processing. Explants were washed three times with fresh, sterile media, in order to remove blood and mucous when was present. Individual basal cells were then isolated by scraping. Isolated basal cells were seeded and expanded in culture flasks in a humidified incubator at 5% $CO_2$ and 37 °C, using the Pneumacult-Ex Plus medium (STEMCELL Technologies) supplemented with 1 mg/mL primozin, 100 μg/mL gentamycin and 2.5 μg/mL amphotericin B. Following expansion, $1.5 \times 10^5$ cells were seeded onto 6.5-diameter-size transwells with 0.4 μm pore polyester membrane inserts (Corning Incorporated) previously coated with type I collagen (Gibco). When cells reached confluency, ALI stablishment, differentiation and maintenance of the cells was accomplished as described above for BCi-NS1.1 cultures.

## Infection procedure and characterization

Overnight *P. aeruginosa* cultures were used to inoculate 4 mL of LB medium to an $OD_{600nm}$ of 0.01 and grown until mid-exponential phase. Subsequently, 1 mL of culture was centrifuged at $4500 \times g$ for 5 min and bacterial pellets were washed with PBS and resuspended in PBS at a concentration of $10^5$ CFUs/mL. 10 μL of solution, containing $10^3$ CFUs, were inoculated onto the apical side of fully differentiated BCi-NS1.1 or CF primary cell ALI cultures which had 600 μL of supplemented Pneumacult-ALI maintenance medium in the basolateral chamber. The same volume of bacteria-free PBS was added to control wells. Infected cells were incubated at 37 °C and 5% $CO_2$ in a humidified incubator for the desired number of hours for the infection to be characterized, after which 200 μL of PBS were added to the apical side and TEER was measured.

The 200 μL of PBS from the apical side and the 600 μL of medium of the basolateral side were harvested, and CFUs present in these solutions were determined by plating 10 μL of serial dilutions on LB-agar plates. The remaining samples were stored at −80 °C for later use. At this point, the cells were destined for counting bacterial CFUs attached to the epithelium or for visualization by confocal microscopy. In the first case, 200 μL of PBS were added to the apical part and the epithelia was disrupted by using a cell scraper. CFUs present in the resulting solution were determined by LB-agar plating of 10 μL in serial dilutions. The absence of bacterial aggregates in the bacterial suspensions prior to plating was validated by visualization in a Leica DM4000 B epifluorescence microscope.

For preparing infected ALI cultures for confocal microscopy, samples were rinsed once with PBS and fixed with 4% paraformaldehyde, both added in the apical and the basolateral chambers, for 20 min at 4 °C. After fixation, cells were washed 3 times with PBS and permeabilized and blocked by incubation with a buffer containing 3% BSA, 1% Saponin and 1% Triton X-100 in PBS for 1 h. A solution of 100 μL containing Phalloidin-AF555 (Invitrogen) and TO-PRO3 (Biolegend) diluted 1:400 and 1:1000, respectively, in a staining buffer (3% BSA and 1% Saponin in PBS), was added to the apical chamber of the cells, which were incubated for 2 h at room temperature for staining. Transwells with fixed and stained infected epithelia were removed from their supports with a scalpel and mounted on glass slides with VECTA-SHIELD® Antifade Mounting Medium (VWR). Microscopy slides of infected cultures were imaged with a Leica Stellaris 8 Confocal Microscope (40× magnification, 1.3 oil) and analyzed using the LasX software 1.4.4.26810 (Leica) for infections on BCi-NS1.1 cells, and with a Carl Zeiss LSM 510 Confocal Laser Scanning Microscope (40× magnification, 1.3 oil) and analyzed using the ZEN software 3.7.97.03000 (Zeiss) for infections on CF primary cells.

The previously harvested basolateral media at the end of the infection was also used for measuring LDH and IL-8 release, following the manufacturer´s instructions of the Invitrogen™ CyQUANT™ LDH Cytotoxicity Assay Kit (Invitrogen) and Human IL-8/CXCL8 DuoSet ELISA Kit (R&D Systems), respectively.

For exploring the effectiveness of antibiotics against bacteria colonizing the epithelium of the infection system, first, an infection was started as described above. After 7 h of incubation at 37 °C and 5% $CO_2$ in a humidified incubator, antibiotics were added in the basolateral medium and the system was further incubated at 37 °C and 5% $CO_2$ in a humidified incubator during 12 h. Afterwards, TEER was measured and CFUs present in apical side, basolateral medium or attached to the epithelium were determined by plating 10 μL of serial dilutions of the corresponding harvested bacterial suspension on LB-agar plates as described above. For this experiment, we analyzed the results corresponding to the antibiotic concentration that reduced the growth of the wild-type strain while not completely inhibiting it in the infection system. This approach was chosen to facilitate a comparative assessment of the protection from antibiotics that the different strains presented under the infection system conditions.

## RNA preparation, RNA-sequencing and RT-qPCR

Twenty mL of LB medium were inoculated to a final $OD_{600nm}$ of 0.01 with different bacterial strains departing from their respective overnight cultures. The cultures were grown until mid-exponential phase and cells were harvested by centrifugation at $5700 \times g$ at 4 °C for 20 min. Cells from the pellets were disrupted by 10 min incubation with 1 mg/mL lysozyme and sonication for 30 s at 0.45 mΩ twice[56]. RNA from the intracellular content was extracted by using the rNeasy minikit (Qiagen), following supplier's instructions. DNA was eliminated by a dNase I (Qiagen) treatment and a subsequent Turbo DNA-free (Invitrogen) treatment. The absence of DNA contamination was checked by PCR amplification of the housekeeping gene *rpsL* using primers *rpsL*_Fw and *rpsL*_Rv (Supplementary Table 1).

RNA-sequencing of pooled RNAs from three independent cultures of each strain[53] was accomplished at the Next Generation Sequencing Service of the Centre for Research in Agricultural Genomics, Barcelona, Spain. Sequencing was performed in an Ion PGM™ Sequencer by using paired end format lectures (2 × 75 bp). Sequences were aligned against the PAO1 reference genome NC_002516.1[57]) and the numeric value of gene expression was normalized to Reads Per Kilobase of gene per million Mapped reads (RPKM) by using the CLC Genomics Workbench software 9.0 (QIAGEN). For minimizing misleading fold change values due to RPKMs close to 0, a cut-off value of 1 was added to each RPKM, as previously suggested[58]. Subsequently, fold change of

RPKM + 1 of the mutant respect to its parental strain PAO1, and the $\log_2$ fold change, were calculated. Genes presenting a $\log_2$ fold change ≤−1 or ≥1 were considered to be relevantly affected in their expression and grouped in the functional classes stablished in PseudoCAP[57]. Transcriptomic data included in this work are deposited in SRA database with accession code PRJNA990706.

RT-qPCR was performed with primers at 400 nM and 50 ng of cDNA from each sample. cDNA was obtained from 10 µg of RNA following the supplier´s instruction of the High-Capacity cDNA reverse transcription kit (Applied Biosystems). The used oligonucleotides (Supplementary Table 1) were designed with Primer3 Input software 4.1.0 and their efficiency was checked by performing a RT-qPCR with serial dilutions of cDNA. The reactions were carried out with Power SYBR green PCR master mix (applied Biosystems), following supplier's instructions, using 96-well plates in an ABI Prism 7500 Realtime PCR system (Applied Biosystems). The quantification of the amplification of the housekeeping gene *rplU*, performed with primers *rplU*_Fw and *rplU*_Rv (Supplementary Table 1), was used as a reference. The analysis of differences in relative quantities of mRNA was performed according to the $2^{-\Delta\Delta CT}$ method[59,60]. The average of three technical replicates of each of three independent biological replicates was used to determine the relative mRNA expression in all cases.

## MIC determination
The MIC of ciprofloxacin, ceftazidime or tobramycin of diverse *P. aeruginosa* strains was determined by broth microdilution method in MH medium, Pneumacult-ALI maintenance medium or SCFM. SCFM contains 66.6 mM $Na^+$, 15.8 mM $K^+$, 2.3 mM $NH_4^+$, 1.7 $Ca^{2+}$, 0.6 $Mg^{2+}$, 79.1 $Cl^-$, 0.35 mM $NO_3^-$, 2.5 mM $PO_4^{2-}$, 0.27 mM $SO_4^{2-}$, 1.4 mM Serine, 1 mM Threonine, 1.8 mM Alanine, 1.2 mM Glycine, 1.7 mM Proline, 1.1 mM Isoleucine, 1.6 mM Leucine, 1.1 mM Valine, 0.8 mM Aspartate, 1.5 mM Glutamate, 0.5 mM Phenylalanine, 0.8 mM Tyrosine, 0.01 mM Tryptophan, 2.1 mM Lysine, 0.5 mM Histidine, 0.3 mM Arginine, 0.7 mM Ornithine, 0.2 mM Cysteine, 0.6 mM Methionine, 3.2 mM Glucose, 9 mM Lactate and 3.6 µM $FeSO_4$[61]. Bacterial CFUs present at the antibiotic concentration corresponding to half of the MIC of the wild-type PAO1 strain measured in MH were determined by plating 10 µL of serial dilutions on LB-agar plates in technical triplicates.

## Reporting summary
Further information on research design is available in the Nature Portfolio Reporting Summary linked to this article.

# Data availability
All data necessary for supporting the findings of this study are enclosed in this manuscript and its associated data indicated here. Raw data generated in this study are provided in the Source Data file. Transcriptomic data included in this work are deposited in SRA database with accession code PRJNA990706. The PAO1 refence genome used during transcriptomic analysis can be found in GenBank database with the access code NC_002516.1. During transcriptomic analysis, genes were grouped in the functional classes stablished in PseudoCAP [https://pseudomonas.com/pseudocap]. Source data are provided with this paper.

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

## Acknowledgements

We thank Professor Ronald G. Cristal (Weil Cornell Medical College, New York, USA) for the Basal Cell Immortalized Non-Smoker 1.1 (BCi-NS1.1) cell line and Professor Víctor de Lorenzo (Centro Nacional de Biotecnología, CSIC, Madrid, Spain) for the donation of the pSEVA234 plasmid. P.L. was the recipient of an European Molecular Biology Organization (EMBO) Scientific Exchange Grant (Ref. nr: 9112) and afterwards a of an ERS/EU RESPIRE4 Marie Skłodowska-Curie Post-doctoral Research Fellowship (Ref. nr: R4202305-01047; this project has received funding from the European Respiratory Society and the European Union's H2020 research and innovation programme under the Marie Skłodowska-Curie grant agreement No 847462). Work in J.L.M. laboratory was supported by the Ministerio de Ciencia e Innovación/Agencia Estatal de Investigación (MCIN/AEI) 10.13039/501100011033 through grant PID2020-113521RB-I00. The work was further supported by a Novo Nordisk Foundation Challenge grant NNF19OC0056411 and a grant from THE JOHN AND BIRTHE MEYER FOUNDATION (2022) to H.K.J.

## Author contributions

P.L., S.H.A., J.L.M., S.M., and H.K.J. designed the project. P.L. and S.L. performed the experiments of characterization of bacterial strains in the infection system, M.A.R. constructed the *mexZ\** mutant and performed the transcriptomic analysis, P.L. constructed the rest of the mutants used for this work and performed gene expression assays, and P.L. and S.L. analyzed the results of the experiments. K.A. obtained the patient samples subsequently used to produce infection model cell cultures. P.L. wrote the first draft of the paper, which was afterwards reviewed by all authors. All authors approved the final version of the manuscript.

## Competing interests

The authors declare no competing interests.
