## [Peer Review File · Nature Communications]

Mutations in the efflux pump regulator MexZ shift tissue colonization by *Pseudomonas aeruginosa* to a state of antibiotic toleranceReviewer #1 (Remarks to the Author):

In this manuscript the authors investigate the role of a *mexZ* inactivating mutation in *Pseudomonas aeruginosa* in epithelial cell colonization. Using transcriptomics the authors find that *lecA* lectin is upregulated in the *mexZ* mutant and go on to show that the *mexZ* mutant invades epithelial cells in a polarized air liquid interface epithelial cell model. Using genetics, the authors show that increased *mexZ* epithelial invasion is dependent upon *lecA* and *lecA* expression can be restored to wild-type levels by either deleting *mexY* or overexpressing *oprM*. Furthermore, deleting *mexB* also increased *lecA* expression. All together these data support a model whereby *mexZ* mutants lead to over expression of *mexXY* which sequester *oprM* away from *mexAB*, causing reduced *mexAB* activity and increasing *mexAB*-dependent quorum sensing gene expression. Finally, using antibiotic killing assays during epithelial cell infections, the authors show that the *mexZ* mutant is more resistant to ceftazidime and tobramycin during epithelial cell infection. All together these data suggest that *mexZ* mutations may benefit *P. aeruginosa* by promoting intracellular invasion which would protect these bacteria from killing by antibiotics that fail to penetrate epithelial barriers. Some missing MIC data would help clarify some of the results, but otherwise few weaknesses are noted. Altogether this is an interesting study that provides new insights into the role of *mexZ* mutants and why they may be under selection during chronic cystic fibrosis infections.

Major comments:

Are the ALI cultures made with wild type or CF epithelial cells? This is an important distinction because CF epithelial cells have been shown to be more susceptible to intracellular invasion. Since the *mexZ* mutations are found frequently in CF isolates, the type of cells used for the experiments are important to consider.

It would be helpful to present in vitro MIC data for tobramycin, ceftazidime, and ciprofloxacin for all the *P. aeruginosa* mutant strains used in the study for cell infections and gene expression analyses. This could be done with a table. This would also help verify that the mutants are properly constructed, based on expected antibiotic susceptibilities for the different efflux pump mutants. These data would also help confirm phenotypes since complementation strains were not made for the *mexZ** mutant or others.

In the discussion, it would be interesting to consider how selection for strains that invade epithelial cells might impact cystic fibrosis lung disease. There is much ongoing controversy regarding the roles of intracellular *P. aeruginosa*, especially in CF infections. These data support the notion that the intracellular niche is under selection because *mexZ* mutants are highly prevalent and clearly increase intracellular invasion by *P. aeruginosa*.

It would also be interesting to consider how *lasR* mutations could affect the phenotypes of *mexZ* mutants. In the previous study were *mexZ* and *lasR* mutations ever detected in the same strain? If so, would you expect *lasR mexZ* double mutants to still invade more, or would they no longer invade due to reduced quorum sensing activity?

Reviewer #2 (Remarks to the Author):

The manuscript is well written and well discussed. The authors could consider few things listed below to be discussed or addressed

Things to be noted :

MexZ mutations are very common in CF isolates were as found less in other common infections.

The authors have tested their MICs in the MH broth which nowhere matches to the conditions in

host. The same hypothesis could be tested in different medium (Artificial CF medium or so) and see if their results make any changes.

Similarly, the amount of oxygen the bacteria gets is far different in lab grown condition to host. Which means an oxygen deprived condition along with medium mimicking host conditions could lead to mexZ mutations (different studies have partly shown).

Why not the author use some clinically relevant mutations of mexZ and test their hypothesis to strengthen their results. For example; frequently observed clinical mutants of mexZ

Further if mexZ mutations alter QS signaling, it may then also impact on biofilm production which may help the adhesion of mexZ mutants when compared to lab strain PAO1

Reviewer #3 (Remarks to the Author):

- What are the noteworthy results?

The observation that mexZ mutations (which are commonly seen in CF isolates) change the colonization behavior of Pa is noteworthy because a) these mutations are common b) the interaction between the efflux pump systems, QS and virulence isn't very well understood and is a fundamental biologic question.

- Will the work be of significance to the field and related fields? How does it compare to the established literature? If the work is not original, please provide relevant references.

The work is original (to my knowledge). The work is significant as follows: a) it may lead to further studies in the role of efflux pumps during physiologic growth conditions b) it also shows another mechanism for antimicrobial tolerance (location of the bugs in the tight junction space) due to mexZ mutations

- Does the work support the conclusions and claims, or is additional evidence needed? Are there any flaws in the data analysis, interpretation and conclusions? Do these prohibit publication or require revision?

Major Comment

- The main weakness of the work (which does not necessarily mean that it should not be published) is the absence of human histopathology showing bacterial localization along the tight junctions in the epithelium in the lung. Most of the published pathology focuses on bacterial aggregates suspended in sputum or around the airway surface liquid. The general experimental plan is logical and well done. The confocal imaging results in Fig 1C (and other figures) are striking. Genetic (but not biochemical) evidence is shown to support the competition for OprM as a mechanism – which is reasonable in a short publication.

Minor Comments

- The fold-change in attached bacteria in Figure 1B is modest (2-fold), and PsA (especially as a biofilm) can be very clumpy. Please show that the attached bacteria are in a single cell suspension for both genotypes prior to plating for CFU counting.
- Figure 1B Suggests a change in overall growth rate between the genotypes under these conditions as the total number of recovered CFU varies by genotype. Please discuss further.
- Please report the absolute values for TEER in Figure S1
- Line 124 - My understanding is LDH is released when the plasma membrane is breached, so you are not directly measuring tight junction function or paracellular permeability here, but rather epithelial viability.
- The manuscript might be easier to read if you include a Supp Figure summarizing the QS related gene expression changes. Otherwise readers will need to download the excel, sort columns etc.

- I couldn't find descriptions of the antibiotic exposures in Figure 3 in the Methods section. If missing, please elaborate.
- The tobramycin concentration used in Fig 3 is lower than concentrations obtained through inhalation (usual delivery route). It is also hard to tell how killing dynamics are changed because only a single drug dose is shown (with no discussion of if these are the in vivo concentrations seen in the airway lumen). For tobramycin, please provide a at least one higher concentration closer to the airway concentration (<https://www.ncbi.nlm.nih.gov/pmc/articles/PMC3621259/>). The results around 0.5 MIC (from liquid culture) suggest that in this particular model the high biofilm tolerance to antibiotics seen in other co-culture models isn't in play which makes it harder to extrapolate the data to human pathophysiology
- The y-axis in Fig B one is difficult to read (and similar graphs through the paper). Consider showing using scientific notation.
- Show absolute values for TEER in Fig 3B as well
- Line 236-238. Were these experiments run as competition assays? From the methods, it looks like the susceptibility testing was done in mono-culture. Consider alternative wording.
-

REVIEWER COMMENTS

We find it important to emphasize to all reviewers that the overall purpose of this investigation was to resolve an important paradox observed in previous communications: the apparent strong selection for *mexZ* mutations in *Pseudomonas aeruginosa* infecting CF airways, despite the marginal increase in antibiotic resistance of these variants (see Frimodt-Møller J, et al., 2018, Sci Rep. doi: 10.1038/s41598-018-30972-y). We have approached this paradox from a biological perspective in this communication, and we have discovered a biological mechanism, which satisfactorily and surprisingly explains how *mexZ*-mutated bacteria are able to persist in the lungs of antibiotic treated patients - despite their low level of clinical antibiotic resistance.

Reviewer #1 (Remarks to the Author):

In this manuscript the authors investigate the role of a *mexZ* inactivating mutation in *Pseudomonas aeruginosa* in epithelial cell colonization. Using transcriptomics the authors find that *lecA* lectin is upregulated in the *mexZ* mutant and go on to show that the *mexZ* mutant invades epithelial cells in a polarized air liquid interface epithelial cell model. Using genetics, the authors show that increased *mexZ* epithelial invasion is dependent upon *lecA* and *lecA* expression can be restored to wild-type levels by either deleting *mexY* or overexpressing *oprM*. Furthermore, deleting *mexB* also increased *lecA* expression. All together these data support a model whereby *mexZ* mutants lead to over expression of *mexXY* which sequester *oprM* away from *mexAB*, causing reduced *mexAB* activity and increasing *mexAB*-dependent quorum sensing gene expression. Finally, using antibiotic killing assays during epithelial cell infections, the authors show that the *mexZ* mutant is more resistant to ceftazidime and tobramycin during epithelial cell infection. All together these data suggest that *mexZ* mutations may benefit *P. aeruginosa* by promoting intracellular invasion which would protect these bacteria from killing by antibiotics that fail to penetrate epithelial barriers. Some missing MIC data would help clarify some of the results, but otherwise few weaknesses are noted. Altogether this is an interesting study that provides new insights into the role of *mexZ* mutants and why they may be under selection during chronic cystic fibrosis infections.

Major comments:

Are the ALI cultures made with wild type or CF epithelial cells? This is an important distinction because CF epithelial cells have been shown to be more susceptible to intracellular invasion. Since the *mexZ* mutations are found frequently in CF isolates, the type of cells used for the experiments are important to consider.

Answer: As part of the optimization process for this infection system, various types of cell cultures were assessed, including cultures derived from primary human cells isolated from people with CF. The colonization behavior of *P. aeruginosa* in CF cell cultures provided the first evidence for tissue invasion by the *mexZ* mutant bacteria in contrast to the surface location of wild-type bacteria. It is important to note, though, that these primary cells can only be passaged a limited number of times before losing their differentiation capacity, making necessary a recurrent patient sampling. Given the availability of a cell line (BCi-NS1.1) that provides a fully differentiated pseudostratified epithelium and does not significantly affect the results concerning bacterial behavior, we opted to utilize the BCi-NS1.1 cell line to characterize the particular *mexZ* mutant bacterial colonization phenotype as well as its causes and consequences. The results from the CF cell cultures are now included in the manuscript (Figure 1A), as well as an explanation for the choice of the BCi-NS1.1 cells for all subsequent experiments (lines 121-138 and 491-504).

It would be helpful to present in vitro MIC data for tobramycin, ceftazidime, and ciprofloxacin for all the *P. aeruginosa* mutant strains used in the study for cell infections and gene expression analyses. This

could be done with a table. This would also help verify that the mutants are properly constructed, based on expected antibiotic susceptibilities for the different efflux pump mutants. These data would also help confirm phenotypes since complementation strains were not made for the *mexZ** mutant or others.

Answer: In response to Reviewer 1's suggestion, we have now included a table in the supplementary material (Supplementary Table 3), presenting the *in vitro* MIC data for tobramycin, ceftazidime, and ciprofloxacin for all *P. aeruginosa* mutant strains used in this study. The results confirm that the mutant strains exhibit the expected antibiotic susceptibilities, consistent with their efflux pump alterations, further supporting the validity of these mutants.

In the discussion, it would be interesting to consider how selection for strains that invade epithelial cells might impact cystic fibrosis lung disease. There is much ongoing controversy regarding the roles of intracellular *P. aeruginosa*, especially in CF infections. These data support the notion that the intracellular niche is under selection because *mexZ* mutants are highly prevalent and clearly increase intracellular invasion by *P. aeruginosa*.

Answer: The limited information on the intraepithelial lifestyle of *P. aeruginosa* may, in part, stem from the limitations of samples from infected patients. Obtaining CF lung tissue samples with sufficient material for in-depth analysis of infecting microorganisms is uncommon. These samples usually originate from lung explants, a source that has become less prevalent due to advancements in CF patient treatment and management (by modulator therapy). Moreover, lung explants are derived from end-stage CF lung disease patients, a condition that may not faithfully represent most CF individuals with chronic infections (see Malet K, et al. BioRxiv, 2023, <https://doi.org/10.1101/2023.08.17.552973>). By using our infection model system, we were able to overcome these challenges and investigate the phenotypic differences of different strains under controlled conditions mimicking as much as possible the conditions in the infected lung. It is important to note here that, as shown in the manuscript figures, we observed bacteria accumulating in the intraepithelial compartment, surrounding the human cells, but not intracellularly, something for which CF epithelia were also observed to be more susceptible (see Sajjan U, et al., 2004, *Infect Immun.* doi: 10.1128/IAI.72.7.4188-4199.2004). According to our findings, mutations in *mexZ* seem to induce this distinct bacterial behavior that could enhance bacterial recalcitrance during infection by providing increased protection against some antibiotics. The whole implications of this behavior besides the increased bacterial recalcitrance (i.e., possible advantages of the phenotype against immune cells or facilitating access to a nutrient niche distinct from the mucus layer), is far from the objective of this article and remains to be deeply elucidated in future works. As suggested, we have expanded our discussion on these issues, since we also believe that our study may inspire further research (lines 358-369).

It would also be interesting to consider how *lasR* mutations could affect the phenotypes of *mexZ* mutants. In the previous study were *mexZ* and *lasR* mutations ever detected in the same strain? If so, would you expect *lasR mexZ* double mutants to still invade more, or would they no longer invade due to reduced quorum sensing activity?

Answer: By analyzing the mutations detected in the clinical strain collection from the Copenhagen CF Centre at Rigshospitalet (see Marvig R, et al. *Nat Genet* 2015. doi: 10.1038/ng.3148), we found that 193 strains had acquired mutations in *mexZ* during the course of the infection, while mutations in *lasR* were acquired in 50 clinical strains of the collection. This made *mexZ* the most frequently mutated gene, and *lasR* was ranked 22nd in frequency of mutation. The fact that these 2 genes are among those frequently mutated may explain why there are a few random clinical strains with both genes mutated (20 strains). We do not consider this rare combination of the mutations as an indicator of special selection and enrichment. Besides, it is important to highlight that published information points to a

complex regulation of quorum sensing in *P. aeruginosa*, which involves more actors and not only LasR (see Lee, et al., 2013, Nat. Chem Biol, doi: 10.1038/nchembio.1225; and Groleau, et al., 2020, mSystems, doi: 10.1128/mSystems.00194-20). Furthermore, it is worth noting that the principal regulators of *lecA* are RhlR and PqsE. Therefore, any influence exerted by LasR on *lecA* expression would likely occur indirectly. Considering this, the impact of a LasR mutation on *lecA* expression, particularly in a *mexZ* mutant already displaying unique *lecA* expression patterns, might be obscured by compensatory mechanisms within the QS regulatory network through RhlR and PqsE.

The significance of exploring this aspect in future research is now highlighted in the manuscript's discussion (lines 393-397).

Reviewer #2 (Remarks to the Author):

The manuscript is well written and well discussed. The authors could consider few things listed below to be discussed or addressed

Things to be noted :

MexZ mutations are very common in CF isolates were as found less in other common infections.

Answer: As highlighted by Reviewer 2, mutations in *mexZ* are very prevalent in CF isolates. The behavior linked to mutations in *mexZ*, elucidated in this study, might contribute to this observed pattern. In fact, as pointed out by Reviewer 1 in one of the comments, the susceptibility of epithelia formed by CF cells to bacterial invasion is higher (see Sajjan U, et al., 2004, *Infect Immun.* doi: 10.1128/IAI.72.7.4188-4199.2004), which could be favoring in some extent the selection of mutations in *mexZ*. Following the comments from Reviewers 1 and 2, we have now incorporated a statement showing this appreciation into the manuscript, enriching the depth of its discussion (lines 356-359).

On the other hand, mutations in *mexZ* are less frequent in other infections, such as *P. aeruginosa* infecting patients with Chronic Obstructive Pulmonary Disease (COPD) (see Eklöf J, et al.. *Clin Microbiol Infect.* 2022 doi: 10.1016/j.cmi.2022.01.017.). In this case, the extensive tissue destruction within the lungs of COPD patients may create an environment where a heightened capacity for bacterial tissue invasion, conferred by *mexZ* mutations, is not as advantageous.

The authors have tested their MICs in the MH broth which nowhere matches to the conditions in host. The same hypothesis could be tested in different medium (Artificial CF medium or so) and see if their results make any changes.

Answer: As pointed out by Reviewer 2, the MICs of the mutants used in this work were performed in MH, the standard methodology used in the field to measure sensitivity to antibiotics. As reviewer 2 stated, MICs may be dependent on the nutrient availability and media composition, as we have previously described (see Laborda P, et al *Microbiol Spectr.* 2022. doi: 10.1128/spectrum.00247-22). Following Reviewer 2's observation, a table with the MICs of all the mutants used in this study measured in different media, including SCFM that mimics the nutritional conditions encountered during infection, are now included in the supplementary material (Supplementary Table 3). As indicated in the supplementary table, while MICs do vary in different media, the differences between mutants and the wild-type strain remain consistent and, more importantly, this new data confirms that the mutant strains exhibit the expected antibiotic susceptibilities, consistent with their efflux pump alterations, further supporting the validity of the mutants used. This information is now included in the results section of the manuscript (lines 279-285).

Similarly, the amount of oxygen the bacteria gets is far different in lab grown condition to host. Which means an oxygen deprived condition along with medium mimicking host conditions could lead to *mexZ* mutations (different studies have partly shown).

Answer: Reviewer 2's observation regarding oxygen levels is entirely valid since oxygen availability differs between infection conditions and laboratory settings. It is worth noting that mutations in *mexZ* may indeed provide advantages under varying oxygen conditions encountered during infection, which presents an intriguing avenue for exploration. This is one of the parameters that this infection system, in its current stage, does not allow to consider. Our laboratory is actively investigating the impact of *mexZ* mutations during bacterial infections in diverse contexts, including this one and their influence

on protection against immune cells. Following Reviewer's comment, we have now incorporated this discussion outlining future avenues for a more comprehensive exploration of the implications of *mexZ* mutations in the context of infection within the manuscript (lines 358-369).

Why not the author use some clinically relevant mutations of *mexZ* and test their hypothesis to strengthen their results. For example; frequently observed clinical mutants of *mexZ*

Answer: It is worth noting that mutations in the *mexZ* gene found in clinical strains are diverse, *but a common trend is their frequent involvement in the C-terminal region of the protein* (see Marvig R, et al. Nat Genet 2015. doi: 10.1038/ng.3148). For this reason, we opted to engineer a mutation that targeted this frequently affected region in our study. In order to obtain clearcut information about the bacterial phenotype during infection we decided to design a mutant strain of PAO1 with a total loss of the mentioned region of MexZ.

This has been further clarified in the manuscript to address any potential concerns of future readers (lines 97-112).

Further if *mexZ* mutations alter QS signaling, it may then also impact on biofilm production which may help the adhesion of *mexZ* mutants when compared to lab strain PAO1

Answer: As noted by Reviewer 2, the QS system plays a significant role in regulating various virulence factors of *P. aeruginosa*, including biofilm formation. We have conducted measurements of an estimation of biofilm formation capacity in the mutants generated during this work. This was performed by measuring the bacterial aggregation to peg lids, as previously described (see Bartell JA, et al., 2019, Nat Commun, doi: 10.1038/s41467-019-08504-7.) In general, we observed only minor differences in biofilm formation estimation when comparing the mutants to PAO1 wild type strain, as shown in the image below.

Estimated biofilm formation capacity by the mutants used during this work. Fold changes of biofilm formation estimation of *mexZ**, *mexZ** Δ*lecA*, *mexZ** Δ*mexY* and Δ*mexB* mutant strains respect to the wild-type strain PAO1 are represented. Error bars indicate the standard deviation of the result of analyzing twelve technical replicates. Measured values of OD_{590nm} used for estimating the biofilm

formation capacity of each strain were the following: PAO1: 1.38 ± 0.22 ; *mexZ**: 1.5 ± 0.19 ; *mexZ** Δ *lecA*: 1.17 ± 0.27 ; *mexZ** Δ *mexY*: 1.28 ± 0.21 ; Δ *mexB*: 1.17 ± 0.25 .

In this regard, it is important to acknowledge the intricate interplay between different regulatory pathways within the QS system and the modulation of virulence factors, which depends on several variables (see Dekimpe et al., 2009, Microbiology, doi: 10.1099/mic.0.022764-0; and Wang, et al., 2018, Sci Rep., doi: 10.1038/s41598-018-30813-y). Consequently, alterations in the extrusion of certain QS molecules or their precursors might lead to difficult to predict changes in the regulation of virulence factors. This complexity has been observed in mutants overexpressing various efflux pumps of *P. aeruginosa* involved in the extrusion of QS molecules (see Alcalde-Rico M, et al., 2018, Front Microbiol, doi: 10.3389/fmicb.2018.02752; Alcalde-Rico M, et al., 2020, Environ Microbiol, doi: 10.1111/1462-2920.15177; Muñoz-Cazalla A, et al., 2023, Microb Biotechnol, 2023, doi: 10.1111/1751-7915.14252).

On the other hand, LecA has been described to have a role in biofilm formation of *P. aeruginosa*. However, its function was elucidated to play a role in the maturation phase rather than the initial attachment to the substratum (see Diggle SP, et al., 2006 Environ Microbiol, doi: 10.1111/j.1462-2920.2006.001001.x). Given that we estimated the biofilm formation capacity through bacterial aggregation, as is conventionally practiced, no changes were observed in this parameter.

Reviewer #3 (Remarks to the Author):

- What are the noteworthy results?

The observation that *mexZ* mutations (which are commonly seen in CF isolates) change the colonization behavior of *Pa* is noteworthy because a) these mutations are common b) the interaction between the efflux pump systems, QS and virulence isn't very well understood and is a fundamental biologic question.

- Will the work be of significance to the field and related fields? How does it compare to the established literature? If the work is not original, please provide relevant references.

The work is original (to my knowledge). The work is significant as follows: a) it may lead to further studies in the role of efflux pumps during physiologic growth conditions b) it also shows another mechanism for antimicrobial tolerance (location of the bugs in the tight junction space) due to *mexZ* mutations

- Does the work support the conclusions and claims, or is additional evidence needed? Are there any flaws in the data analysis, interpretation and conclusions? Do these prohibit publication or require revision?

Major Comment

- The main weakness of the work (which does not necessarily mean that it should not be published) is the absence of human histopathology showing bacterial localization along the tight junctions in the epithelium in the lung. Most of the published pathology focuses on bacterial aggregates suspended in sputum or around the airway surface liquid. The general experimental plan is logical and well done. The confocal imaging results in Fig 1C (and other figures) are striking. Genetic (but not biochemical) evidence is shown to support the competition for OprM as a mechanism – which is reasonable in a short publication.

Answer: We agree that there is a total absence of human lung histopathology described in this manuscript. However, the whole point of employing a differentiated lung cell culture in our investigations of the behavior of *mexZ* mutant bacteria relative to wild type bacteria (two isogenic strains except for the *mexZ* gene) has been to investigate the phenotypic differences of the two strains under standard controllable conditions mimicking as much as possible the conditions in the infected lung. This clearly contrasts with the standard clinical investigations of the same strains, which could not provide any satisfactory explanations for the apparent strong selection for *mexZ* mutations in the infected CF patients. We therefore argue that we have taken a strong step forward in the analysis, which in fact takes advantage of a model system reflecting the histopathological conditions in the CF lung, and our study under these lung mimicking conditions has provided a convincing and surprising explanation for the selective success of *mexZ* mutant bacteria during treatment with antibiotics in CF patients.

Minor Comments

- The fold-change in attached bacteria in Figure 1B is modest (2-fold), and PsA (especially as a biofilm) can be very clumpy. Please show that the attached bacteria are in a single cell suspension for both genotypes prior to plating for CFU counting.

Answer: We conducted an analysis of cell suspensions by visualization in an epifluorescence microscope prior to performing CFU plating. The images displayed below show the GFP-tagged bacterial suspensions harvested after 14 hours of infection observed under an epifluorescence microscope. These images illustrate that the bacterial cells were in a single cell suspension, indicating

that no clumping occurred under the specified conditions, potentially due to the insufficient bacterial density required for clump formation. We would like to thank Reviewer 3 for raising this possibility, which has prompted us to include a clarification of this validation step in the Materials and Methods section of the manuscript (lines 523-524). This input has greatly contributed to the robustness of our experimental approach.

- Figure 1B Suggests a change in overall growth rate between the genotypes under these conditions as the total number of recovered CFU varies by genotype. Please discuss further.

Answer: Correct: the total bacterial load of the *mexZ** mutant is higher than that of PAO1 without mutation in *mexZ*. Previous studies have shown that disruptions in the *mexZ* gene are not associated with significant changes in bacterial respiration when grown on different carbon or nitrogen sources (see Frimodt-Møller J, et al., Sci Rep, 2018, doi: 10.1038/s41598-018-30972-y).

In the context of our infection system, the growth of apical bacteria, those not attached to the cells, is expected to be supported by the mucus produced by the epithelium, while basolateral bacteria rely on the nutrients present in the basolateral media. Our current hypothesis is that the *mexZ** mutant, due to its tendency to accumulate within the epithelial barrier, may have easier access to the nutrients in that compartment. This could be a niche for nutrients distinct from those in the mucus layer, which might provide the *mexZ** mutant with an additional advantage. In addition, there could be further advantages of the bacterial behavior described in the current work, such as protection against immune cells or improved survival in low-oxygen conditions. In response to the comment, we have expanded our discussion of the further possible approaches in the manuscript (lines 359-369).

- Please report the absolute values for TEER in Figure S1

Answer: A table containing the absolute TEER values is now included in the supplementary material (Supplementary Table 4).

- Line 124 - My understanding is LDH is released when the plasma membrane is breached, so you are not directly measuring tight junction function or paracellular permeability here, but rather epithelial viability.

Answer: We thank Reviewer 3 for highlighting this potential point of confusion. We have revised this sentence for better clarity of the text (lines 150-153).

It is important to emphasize that in this study, we have included both LDH and TEER measurements because they provide distinct information. LDH, as pointed out by Reviewer 3, serves as an indicator of damage to individual epithelial cells. On the other hand, TEER measurements indicate the disruption of the epithelial barrier as a whole. By assessing both these parameters, we aim to distinguish bacterial strains that may disrupt the epithelial barrier without causing harm to individual cells, or vice versa.

- The manuscript might be easier to read if you include a Supp Figure summarizing the QS related gene expression changes. Otherwise readers will need to download the excel, sort columns etc.

Answer: In accordance with the recommendation of Reviewer 3, we have extracted the gene expression changes related to QS from the Supplementary data and presented them in a supplementary table (Supplementary Table 2). To enhance clarity and understanding, we have also incorporated a schematic supplementary figure illustrating the detected changes involved in the mechanism underlying the studied phenotype (Supplementary Figure 3).

- I couldn't find descriptions of the antibiotic exposures in Figure 3 in the Methods section. If missing, please elaborate.

Answer: We thank Reviewer 3 for pointing out this, which has now been clarified accordingly (lines 544-555).

- The tobramycin concentration used in Fig 3 is lower than concentrations obtained through inhalation (usual delivery route). It is also hard to tell how killing dynamics are changed because only a single drug dose is shown (with no discussion of if these are the in vivo concentrations seen in the airway lumen). For tobramycin, please provide a at least one higher concentration closer to the airway concentration (<https://www.ncbi.nlm.nih.gov/pmc/articles/PMC3621259/>). The results around 0.5 MIC (from liquid culture) suggest that in this particular model the high biofilm tolerance to antibiotics seen in other co-culture models isn't in play which makes it harder to extrapolate the data to human pathophysiology

Answer: The primary objective in this part of the communication was to assess the potential impact of the *mexZ* mutation on *P. aeruginosa*'s antibiotic tolerance during infection conditions. When addressing this section of the project, we conducted experiments employing different antibiotic concentrations, and it is important to note that a consistent trend was observed across all conditions. We opted to present the specific concentration featured in the article, since it reduced the growth of the wild-type strain while not completely inhibiting it. This approach was chosen to facilitate a comparative assessment of antibiotic tolerance among the different strains under infection conditions. This has now been clarified in the manuscript (lines 253-254 and lines 544-555).

Following Reviewer 3's suggestion, we have now performed an additional experiment in which higher tobramycin concentrations were used (128, 256 and 512 $\mu\text{g/mL}$). The results revealed that these concentrations completely eradicated both the wild-type and *mexZ** mutant strains. This result points to the information mentioned before: during this study we are not using clinical strains (that may evade antibiotic treatments by further mechanisms mediated by many other mutations) because the goal of this experiment was not to reproduce the antibiotic treatment conditions but to be able to compare the effect of a specific mutation in the bacterial antibiotic tolerance. Following Reviewer 3's comment, this has now been further clarified in the manuscript avoiding potential misinterpretations of future readers (lines 253-254 and lines 544-555).

- The y-axis in Fig B one is difficult to read (and similar graphs through the paper). Consider showing using scientific notation.

Answer: The y-axis of the figures has been changed according to Reviewer 3's comment, which has clarified their comprehensiveness.

- Show absolute values for TEER in Fig 3B as well

Answer: A table containing the absolute TEER values is now included in the supplementary material. Additionally, to further increase the transparency of our study, we have also included LDH, IL-8 and CFU values measured in each infection in the same supplementary table (Supplementary Table 4), as well as the rest of parameters measured during this work in subsequent supplementary tables.

- Line 236-238. Were these experiments run as competition assays? From the methods, it looks like the susceptibility testing was done in mono-culture. Consider alternative wording.

Answer: As noted by Reviewer 3, the experiments were not competition assays. This has now been rephrased for clarification (lines 272-273).

Reviewer #1 (Remarks to the Author):

The authors have done an excellent job responding to the prior critiques.

Reviewer #2 (Remarks to the Author):

I am satisfied with the responses the authors have made. The manuscript looks solid with great discussion.

Reviewer #3 (Remarks to the Author):

The authors responded in a comprehensive and acceptable manner to all of my critiques. I have no further concerns.